

# A robust maximum correntropy forecasting model for time series with outliers

Jing Ren[1] and Wei-Qin Li[2]

[1] College of Computer, Xi'an Aeronautical Institute, Xi'an, Shaanxi Province, China
[2] School of Automation and Information Engineering, Xi'an University of Technology, Xi'an, Shaanxi, China

## ABSTRACT

It is of great significance to develop a robust forecasting method for time series. The reliability and accuracy of the traditional model are reduced because the series is polluted by outliers. The present study proposes a robust maximum correntropy autoregressive (MCAR) forecasting model by examining the case of actual power series of Hanzhong City, Shaanxi province, China. In order to reduce the interference of the outlier, the local similarity between data is measured by the Gaussian kernel width of correlation entropy, and the semi-definite relaxation method is used to solve the parameters in MCAR model. The results show that the MCAR model in comparison with deep learning methods, in terms of the average value of the mean absolute percentage error (MAPE), performed better by 1.63%. It was found that maximum correntropy is helpful for reducing the interference of outliers.

## INTRODUCTION

The forecasting of the time series plays an important role in fields of natural science, social science, industrial engineering, financial science and technology and other fields. For instance, in the power system, important decisions are made on account of the forecasting results, including the generating capacity, the reliability analysis of the scheduling plan. However, under the influence of some factors, the time series has obvious variability and non-stationary (*Dudek, 2016*), and therefore the accurate forecasting is increasingly difficult. It becomes imperative to develop the robust and effective forecasting method with higher accuracy (*Fekri et al., 2021*; *José et al., 2019*; *Kong et al., 2019*).

In the past, some machine learning methods, such as the linear regression model (*Ilic et al., 2021*), the autoregressive integrated moving average (ARIMA) model (*Büyükşahin & Ertekin, 2019*), the exponential smoothing (*De Oliveria & Oliveira, 2018*), the grey model (*Huang, Shen & Liu, 2019*), and the fractal extrapolation model (*Wang et al., 2012*), have been proposed for forecasting of time series. Here, the machine learning method can establish a parameter model and forecast the data in the future according to the time series data. Compared with the traditional regression model, it has higher forecasting performance. Recently, some deep learning models, such as the TCN-based model by incorporating calendar and weather information (*Jiang, 2022*), the FF-ANNs by

Corresponding author
Wei-Qin Li, wqlee@xaut.edu.cn

considering the temperature, weekends and load lags (*Rajbhandari et al., 2021*), and the RNN by BO-PSO optimizing the hyper-parameters (*Li, Zhang & Cai, 2021*), are proposed to improve the load forecasting performance and have higher accuracy than traditional methods.

However, the time series is easily polluted by the random noise and outlier, which seriously reduce the reliability and accuracy of the forecasting model. For the interference of the random noise, the wavelet analysis and Kalman filtering are introduced to preprocess time series (*Quilty & Adamowski, 2018*; *Bashir & El-Hawary, 2009*; *Nóbrega & Oliveira, 2019*). Recently, the empirical mode decomposition (EMD) and its improved method are applied to eliminate the interference and randomness of the sequence at different time scales (*Li & Chang, 2018*). As for the outlier, it is caused by the sensor faults, the equipment failures and other unexpected events, and is generally considered as the data beyond the error threshold (*Dixit et al., 2022*). The deep learning approach (*Munir et al., 2019*), data preprocessing (*Wang et al., 2020*) and online sequential outlier robust extreme learning machine (*Zhang et al., 2019*) are introduced to detect and eliminate the outliers, which are depended on the statistical characteristics of the time series. The disadvantage of the methods is that they need to set the threshold and have high time complexity.

To solve the problem of forecasting time series polluted by outlier, a robust regression model without detecting and eliminating outlier is developed in this article. The forecasting model is applicable to the complex occasions of data contaminated by outlier, especially in actual industrial sites, such as electrical load forecasting, wind farm power forecasting, *etc.* This model is expected to improve the efficiency, accuracy and robustness of time series. This article develops a robust maximum correntropy auto-regression (MCAR) forecasting model. First, the similarity of data is assessed by the Gaussian kernel width of correlation entropy to eliminate the outlier. Then, the semi-quadratic method is presented to the quadratic programming for the nonlinear non-convex programming. Lastly, in order to improve the robustness and accuracy, the half a second type of the conjugate convex function and the semi-definite relaxation (SDR) method are developed to estimate the model parameters.

The rest of the article is organized as follows. Firstly, the robust MCAR method is developed for time series with the outlier in "Methods". Then, the forecasting results are analyzed and performances are evaluated in "Results". Next, the comparison with some state of the art forecasting model and discussion are presented in "Discussion". Moreover, the brief finding is introduced in "Findings". Lastly, the conclusion and future directions are presented in the last section.

## METHODS

### The regression model

According to the linear theory, the auto-regression (AR) model of the series $\hat{Y} = \{\hat{y}_1, \ \hat{y}_2, \cdots \hat{y}_M\}$ is shown in Eq. (1)

$$\hat{y}_t = \beta_1 y_{t-1} + \beta_2 y_{t-2} + \cdots + \beta_N y_{t-N} + e_t \tag{1}$$

where $\hat{y}_t, t = 1, 2, \cdots, M$ is the forecasting load at the current moment $t$, $M$ the number of

group, $Y_{t-i}$ the actual load at the past moment $t - i$, $\beta_i$ the regression parameter, $e_t$ the error, and $N$ the regression order. Therefore, it can be expressed to Eq. (2)

$$
\begin{bmatrix} \hat{y}_1 \\ \hat{y}_2 \\ \vdots \\ \hat{y}_M \end{bmatrix} = \beta_1 \begin{bmatrix} y_0 \\ y_1 \\ \vdots \\ y_{M-1} \end{bmatrix} + \beta_2 \begin{bmatrix} y_{-1} \\ y_0 \\ \vdots \\ y_{M-2} \end{bmatrix} + \cdots + \beta_N \begin{bmatrix} y_{1-N} \\ y_{2-N} \\ \vdots \\ y_{M-N} \end{bmatrix} + \begin{bmatrix} e_1 \\ e_2 \\ \vdots \\ e_M \end{bmatrix} \tag{2}
$$

The least square algorithm is adopted to minimize the sum of the squares of errors (*Vovk et al., 2019*; *Midiliç, 2020*). Let $\Phi = [\beta_1, \beta_2, \cdots, \beta_N]$, $Z = \{z_1, z_2, \cdots, z_M\}$ and $z_t = [y_{t-1}, y_{t-2}, \cdots y_{t-N}]^T$. Equation (3) is the objective function

$$
\min_{\Phi} (\hat{Y} - \Phi Z)^2 \tag{3}
$$

Furthermore, Eq. (3) can also be written to Eq. (4)

$$
\min_{\Phi} \sum_{t=1}^{M} (\hat{y}_t - \Phi z_t)^2 \tag{4}
$$

Actually, some factors, such as the machine failure, human errors, can cause the outlier point in the recording process. Consequently, the accuracy can be reduced by the above method. In this work, we develop a robust regression model.

## MCAR forecasting model

### Correntropy analysis

As shown in Eq. (4), $y_t$ at the current moment $t$ is calculated by the weighted sum of the data of past moments. Actually, because $\hat{Y}$ and $\Phi Z$ are two random variables, the objective function of Eq. (3) is the smallest if they have same statistical distributions.

The correntropy is the similarity measure between two random variables, as shown in Eq. (5)

$$
V_\sigma(\hat{Y}, \Phi Z) = E\big[k_\sigma(\hat{Y} - \Phi Z)\big] \tag{5}
$$

where $E[]$ is expectation. $k_\sigma(\hat{Y} - \Phi Z)$ is the Gaussian kernel as illustrated in Eq. (6)

$$
k_\sigma(\hat{Y} - \Phi Z) = \exp\left(-\frac{(\hat{Y} - \Phi Z)^2}{2\sigma^2}\right) \tag{6}
$$

where $\sigma$ is a random parameter representing the kernel width, which is selected by the density estimates (such as the Silverman specification).

In Eq. (6), the joint probability density cannot be calculated directly. Therefore, the Parzen window is developed to estimate the correntropy of the limited samples

$$\hat{V}_\sigma(\hat{Y}, \Phi Z) = \frac{1}{M} \sum_{t=1}^{M} k_\sigma(\hat{y}_t - \Phi z_t) \tag{7}$$

where $M$ is the number of groups as shown in Eq.(1).

In Eq. (7), the kernel width $\sigma$ controls the window of the correntropy. Here, due to the kernel width constraint and negative exponential term in Gaussian kernel, the contribution of large error value to correlation entropy can be reduced. As a result, the numerical instability caused by large deviation value can be avoided, and the adverse effect of outliers can be effectively reduced.

The above method in this work compares the deviation between the output sample (forecasting value) and the real sample (actual value) in the system one by one under the Gauss kernel function, which is a local optimization method. However, the moment expansion considers the data matrix from the whole situation and calculates the error sequence. When the matrix operation is limited, it needs to construct a new calculation method, which increases the algorithm complexity.

From geometric point of view, in the sample space, the mean square error in the least square method is the 2 norm for distance, considering the second-order statistics of the data signal only, and does not reflect the statistical characteristics of the data (*Liu, Pokharel & Principe, 2007*), which makes the convergence of the least squares estimation worse in non Gaussian environment. Gaussian kernel function contains exponential function, and Taylor series of exponential function is shown in Eq. (8)

$$e^{x^2} = 1 + x^2 + \frac{1}{2!}x^4 + \frac{1}{3!}x^6 + \frac{1}{4!}x^8 + \frac{1}{5!}x^{10} + \cdots \tag{8}$$

where $x$ can be regarded as the element corresponding to the position in the error sequence. From Eq. (8), it can be seen that the correntropy contains the even order distance. When the distance between two points is close, it is equivalent to the distance measured as 2 norm. With the increase of the distance, it is similar to 1 norm, or even eventually tends to 0 norm (*Liang, Wang & Zeng, 2015*).

The correntropy reflects the high-order statistical characteristics and can more accurately evaluate the error between the estimated value and the actual one. Therefore, it can reduce the influence of outliers. In this work, the correlation entropy is introduced into the AR model to enhance the robustness.

## MCAR forecasting model

In the regression model, the mean square error of $\hat{Y}$ at the time $t$ and the historical data $\Phi Z$ is the quadratic function of the convex curve along a straight line $\hat{Y} = \Phi Z$. However, the value away from $\hat{Y} = \Phi Z$ will increase mean error of samples, and make the regression parameters have larger error in MSE.

Traditional mean square error is measured in the way of global similarity. All samples in the joint space contribute significantly to the similarity. However, the correntropy is measured in a local way. Due to locality, the value of the similarity is determined by the

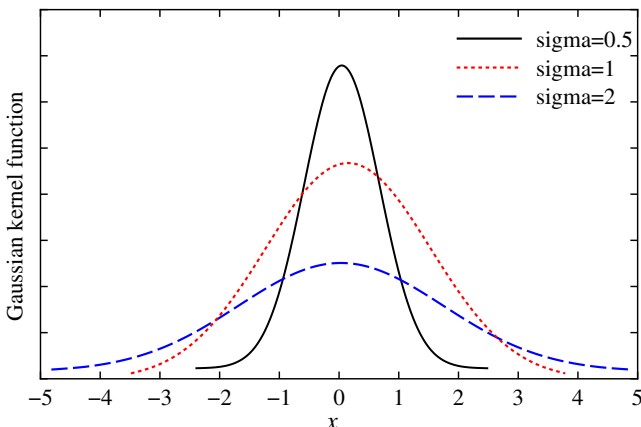

**Figure 1 Gaussian kernel function with different variables.**

kernel function along the line of $\hat{Y} = \Phi Z$ for random variables. Assume that $X = \hat{Y} - \Phi Z$, the Gaussian kernel function of $X$ is shown in the Fig. 1.

From Fig. 1, we can see that if $X = 0$, Gaussian kernel $k_\sigma(\hat{Y} - \Phi Z)$ is maximized, and the residual at the origin is zero. Here, the maximum entropy is a kind of adaptive loss function, known as the maximal entropy criterion. It is suitable for the situation with non-Gaussian and large outlier value (*Santamaria, Pokharel & Principe, 2006*; *Bessa, Miranda & Gama, 2009*; *Chen & Principe, 2012*; *He et al., 2011*). In this work, we establish a robust multidimensional regression model based on the maximal entropy (MC).

Assume that the number of groups and order number of the model are $M$ and $N$, respectively. For the convenience of calculation, the equality constraint $\Phi^T \Phi = 1$ is introduced. Thereby, Eq. (9) is the constraint problem on MC

$$\max_{\Phi} \sum_{t=1}^{M} k_\sigma(\hat{y}_t - \Phi z_t)$$

$$\text{s.t.}\, \Phi^T \Phi = 1 \tag{9}$$

The optimization problem of Eq. (9) is nonlinear and non-convex, and cannot be solved directly. The conjugate convex function is introduced to solve the semi-quadratic form. Here, the auxiliary variables is introduced, and accordingly it can be simplified to Eq. (10)

$$\min_{\Phi} \sum_{t=1}^{M} \omega_t(\hat{y}_t - \Phi z_t)^2$$

$$\text{s.t.}\, \Phi^T \Phi = 1 \tag{10}$$

Actually, the weighted function can reduce the large error term and the adverse effect of the outlier on the optimization result. We define the matrix $R = \text{diag}(w)$, where $w = [\omega_1, \omega_2, \cdots, \omega_M]$, and hence Eq. (10) is equivalent to Eq. (11)

$$\min_{\Phi} R\|\hat{Y} - \Phi Z\|^2$$
$$\text{s.t.} \Phi^T \Phi = 1 \tag{11}$$

The solution of Eq. (11) is a non-convex quadratic programming problem, which is difficult to solve. When the initial condition of $R(0) = diag(1)$, $\Phi$ becomes the optimal solution, and the above Eq. (11) is transformed into a homogeneous constraint programming of Eq. (12)

$$\min_{\Phi} \|\gamma\hat{Y} - \Phi Z\|^2$$
$$\text{s.t.} \gamma^2 = 1, \Phi^T \Phi = 1 \tag{12}$$

It is equivalent to Eq. (13)

$$\min_{\Phi} \begin{bmatrix} \Phi^T & \gamma \end{bmatrix} \begin{bmatrix} Z^T Z & -Z^T \hat{Y} \\ -\hat{Y}^T Z & \|Z\|^2 \end{bmatrix} \begin{bmatrix} \Phi^T \\ \gamma \end{bmatrix}$$
$$\text{s.t.} \gamma^2 = 1, \Phi^T \Phi = 1 \tag{13}$$

Let $\xi = [\Phi^T \gamma]^T$, $B = \begin{bmatrix} I_{N \times N} & 0 \\ 0 & 0 \end{bmatrix}$, and $C = \begin{bmatrix} Z^T Z & -Z^T \hat{Y} \\ -\hat{Y}^T Z & \|Z\|^2 \end{bmatrix}$. Equation (13) can simplified to Eq. (14)

$$\min_{\Phi} \xi^T C \xi$$
$$\text{s.t.} \xi^T B \xi = 1 \tag{14}$$

Using the semi-definite relaxation (SDR) method, the objective function and constraint conditions in Eq. (14) are equivalent to Eq. (15)

$$\xi^T C \xi = \text{Tr}\{\xi^T C \xi\} = \text{Tr}\{C \xi \xi^T\}$$
$$\xi^T B \xi = \text{Tr}\{\xi^T B \xi\} = \text{Tr}\{B \xi \xi^T\} \tag{15}$$

where Tr{} is the trace of the matrix. In Eq. (16), we define the matrix

$$\omega = \xi \xi^T \tag{16}$$

where $\omega$ is a symmetric positive semi-definite (PSD) matrix with a rank of 1. The resulting semi-definite relaxation optimization constraint is shown in Eq. (17)

$$\min_{\Phi} \text{Tr}\{C\omega\}$$
$$\text{s.t.} \text{Tr}\{B\omega\} = 1, \omega \geq 0 \tag{17}$$

Furthermore, let Eq. (18) is the eigen-decomposition of the matrix $\omega$

$$\Phi = VRV^T \tag{18}$$

where $V = [v_1, v_2, \ldots v_M]$ is the eigenvector of $\omega$, and $R = \text{diag}(r_1, r_2 \cdots, r_M)$ is the

corresponding eigenvalue. $\omega(1) = r_1 v_1 v_1{}^T$ is closest to $\omega$ when the rank is 1, and so $\xi$ is estimated in Eq. (19)

$$\xi = \sqrt{r_1 v_1} \tag{19}$$

Finally, the parameter $\Phi$ is the first $N$ value of $\xi$.

In summary, the steps of the robust MCAR forecasting method are as follows:

Step (1): Set the maximum number of iteration, and initialize $R(0) = diag(1)$;

Step (2): Solve the optimization constraint problem in Eq. (17) and $\xi$ by Eq. (19);

Step (3): According to the Silverman specification: $\sigma = 1.06 \times \min(\sigma_e, R/1.34) \times I^{-0.2}$ ($\sigma_e$ is the standard deviation of $\hat{Y} - \Phi Z$, and $R$ is the quartile difference) get solution of $\omega_t$;

Step (4): Perform Steps (2) and Step (3) until the termination condition is reached, and find the regression parameters $\Phi$;

Step (5): Forecast the data at the next moment based on the parameter $\Phi$ in Step (4).

It is worth pointing out that, unlike previous methods (*Bashir & El-Hawary, 2009*), the proposed MCAR model does not need to judge whether there is outliers in the data set, but directly constructs and trains the model based on the data set, since the MCAR model can automatically reduce the impact of outliers by the maximal entropy.

## RESULTS

As a case study, the experimental data is taken from the actual electricity power of Hanzhong City, Shaanxi province, China. Here, the time interval is 1 h. The forecasting is a one-step mode, that is, the current load is forecasted from the historical data of the past $N$ times. To analyze the forecasting performance, the root mean square error (RMSE), the mean absolute error (MAE) and the mean absolute percentage error (MAPE) are employed in Eqs. (20) to (22)

$$RMSE = \sqrt{\frac{\sum_{i=1}^{T}(y_i - \hat{y}_i)^2}{T}} \tag{20}$$

$$MAE = \frac{\sum_{i=1}^{T}|y_i - \hat{y}_i|}{T} \tag{21}$$

$$MAPE = \frac{\sum_{i=1}^{T}\left|\frac{y_i - \hat{y}_i}{y_i}\right|}{T} \times 100\% \tag{22}$$

where $y_i$ and $\hat{y}_i$ represent the actual load value and the forecasting one at the time $t$ respectively, and $T$ is the total forecasting number.

Firstly, the forecasting performance of regression forecasting model is tested. Here, the model order $N$ and group number $M$ on the accuracy are set to 4 and 8, respectively. The solution method is the least square method (LSM). As shown in Fig. 2, the forecasting load curves of 4:00 in 15 days from January 6 to January 20 are shown respectively. It can be seen that, if the load series is slow to change, the forecast error is rather low; contrarily, if the sequence has a greater change (a sharp rise or drop), the forecast accuracy is rather

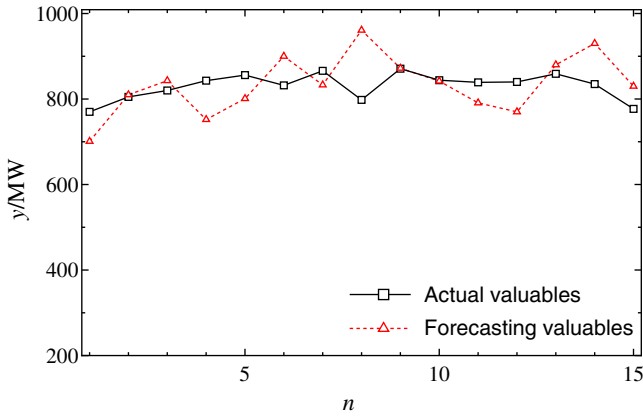

**Figure 2 The forecasting results by regression model at 4:00 from January 6 to 20.** The solid and dot curves indicate that the actual loads and the forecasting results, respectively.

**Table 1 Parameters setting of forecasting models.**

| Parameters | Forecasting models | | |
|---|---|---|---|
| | AR | MDAR | MCAR |
| Solution method | LSM | LSM | LSM |
| Model order | 4 | 4 | 4 |
| Group number | 8 | 8 | 8 |
| Kernel width | – | – | 1,000 |

high. This means that AR forecasting model is sensitive to the sharp change of time series. Here, some effects may lead to load data greater change and regression forecasting model is more sensitive to the outlier.

## Comparison of MCAR and regression models

In this section, the performance of the proposed MCAR model is compared with the differential autoregressive (MDAR) model and the AR model. Here, to guarantee a fair comparison, parameters of three methods are set to the same value as shown in Table 1. The training data sets are the same, which are taken from the January data set. Firstly, the performance of three methods on normal data (no outlier) is verified. Results of the relative errors ($Re$) of the 50 different forecasting points are illustrated in Fig. 3. It can be seen that, the relative error of the AR and the MDAR model have a large error at the forecasting point 7, 29 and 34, due to the sudden increase or decrease of the actual data.

Table 2 shows the corresponding performance indexes. Here, the MAPE of MCAR is 4.74%, while that of the AR and MDAR models are greater than 7%. Meanwhile, the other performance indexes of MCAR are relatively small, indicating that the proposed MCAR model is superior to the other regression models.

Next, the robustness of the MCAR model on the outlier is further verified. Fig. 4 shows the comparison of forecasting results from January 6 (day 1) to January 20 (day 15). It can

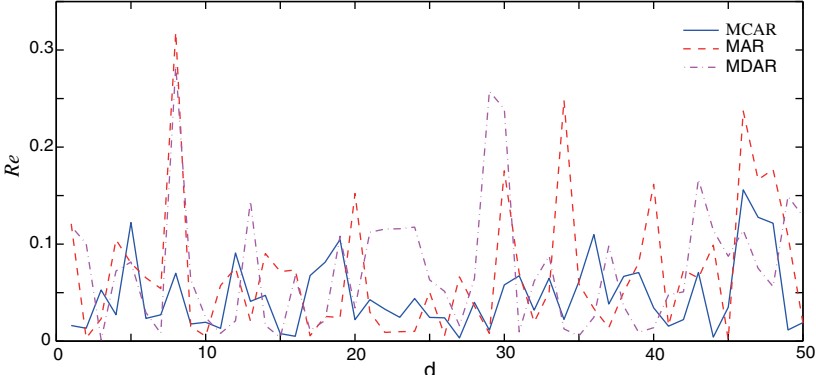

**Figure 3 Comparison of the forecasting relative errors of normal data (no outlier).** The solid, dash, and dot-dash curves are the relative errors of the outlier data of MCAR, MAR, and MDAR, respectively.

**Table 2 Performance indexes of the normal data of MCAR, AR and MDAR models.**

| Performance indexes | Forecasting models | | |
|---|---|---|---|
| | **AR** | **MDAR** | **MCAR** |
| RMSE (MW) | 85.213 | 84.032 | 54.015 |
| MAPE (%) | 7.08 | 7.44 | 4.74 |
| MAE (MW) | 60.727 | 63.695 | 40.964 |

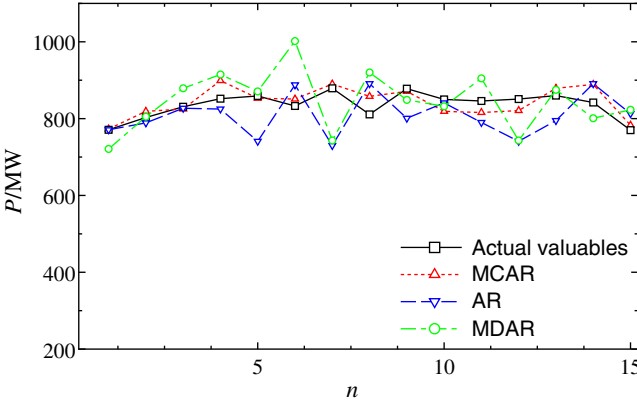

**Figure 4 Comparison of the forecasting results with outlier data at 4:00 from January 6 to 20.** The dot, dash, and dot-dash curves show the forecasting results of MCAR, MAR, and MDAR, respectively.

be observed that, after the training sample with the outlier, the forecasting values of multiple points of AR and MDAR deviate greatly. Moreover, the results of MCAR are still good compared with the actual valuables, which mean that the proposed MCAR model in this work has higher robustness.

The comparisons of the relative errors ($Re$) between the forecasting result and the actual one are shown in Fig. 5. It can be seen that, AR and MDAR models have large forecasting

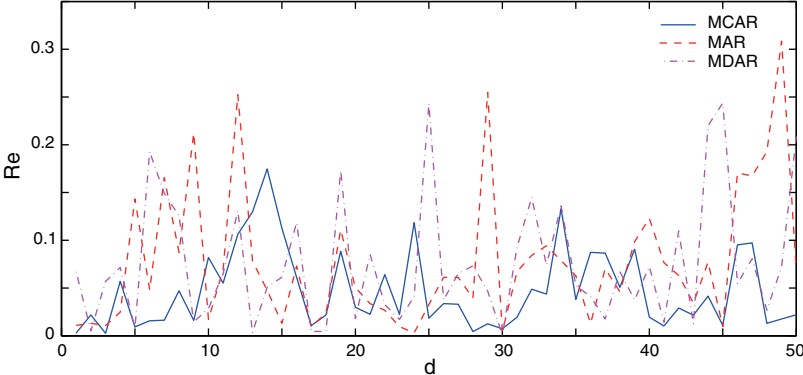

**Figure 5 Comparison of the forecasting relative errors of data with the outlier.** The solid, dash, dot-dash curves are the relative errors of MCAR, MAR, and MDAR, respectively.

**Table 3 Performance indexes of AR, MDAR and MCAR with outlier data.**

| Performance indexes | Forecasting model | | |
|---|---|---|---|
| | AR | MDAR | MCAR |
| RMSE (MW) | 92.783 | 84.818 | 56.655 |
| MAPE (%) | 7.79 | 7.55 | 4.88 |
| MAE (MW) | 67.752 | 64.500 | 41.952 |

errors. The relative error of the MCAR model remains relatively low, indicating that it is less affected by the outlier. From the performance indexes in Table 3, it can be seen that the MAPE of MCAR model is 4.88%, which is significantly smaller than that of the AR and MDAR models. In addition, compared with results with no outlier in Table 2, performance indexes did not increase significantly after the outlier data was added. This shows that, the MCAR model proposed in this work is robust to the outlier and can improve the forecasting accuracy.

## Parameters selection of MCAR

The influences of parameter kernel width $\sigma$, model order $N$ and group number $M$ on the accuracy are discussed.

The influences of parameters of the Gaussian kernel on the forecasting performance are analyzed. Actually, for different data sets, the width of Gaussian kernel is different. Here, the Silverman rule is used to determinate the kernel width. The performance indexes are calculated using the forecasting valuables obtained from each kernel width, compared with the Silverman rule. It can be seen from Table 4 that, RMSE, MAPE and MAE of Silverman rule are smaller than the corresponding valuables with other kernel width. Here, the model considers the standard deviation and quartile potential difference of error sequence synthetically, realizing the restriction of increasing the correntropy to the larger error estimate value, and improves the robustness of the correntropy measurement method to the outlier value.

**Table 4 Performance indexes of MCAR with different kernel sizes.**

| Kernel size | Performance indexes | | |
| --- | --- | --- | --- |
| | RMSE (MW) | MAPE (%) | MAE (MW) |
| 1,000 | 316.704 | 14.25 | 119.954 |
| 1,200 | 317.115 | 14.55 | 122.516 |
| 1,500 | 81.285 | 7.98 | 67.884 |
| 2,000 | 181.499 | 11.71 | 98.329 |
| 2,500 | 102.938 | 10.62 | 91.065 |
| Silverman | 63.058 | 4.18 | 44.412 |

**Table 5 Comparison of performance indexes of different model orders.**

| Model order | Performance indexes (no outliers) | | | Performance indexes (with outliers) | | |
| --- | --- | --- | --- | --- | --- | --- |
| | RMSE (MW) | MAPE (%) | MAE (MW) | RMSE (MW) | MAPE (%) | MAE (MW) |
| 3 | 53.593 | 0.0504 | 42.282 | 63.808 | 0.0598 | 49.889 |
| 4 | 53.499 | 0.0474 | 40.964 | 56.655 | 0.0488 | 41.952 |
| 5 | 86.611 | 0.0693 | 62.569 | 83.532 | 0.0742 | 65.976 |
| 6 | 85.198 | 0.0648 | 67.063 | 90.767 | 0.0644 | 68.483 |

**Note:**
The data indicate the performance indexes of the normal data and the data with the outlier of different model orders. The performance indexes are smallest when the order $N = 4$.

**Table 6 RSSN and AICc values of different model orders.**

| Model order | 3 | 4 | 5 | 6 | 7 |
| --- | --- | --- | --- | --- | --- |
| RSSN | 143.28 | 25.26 | 53.02 | 250.34 | 239.57 |
| AICc | 40.52 | 34.07 | 48.77 | 74.51 | 86.22 |

**Note:**
The data indicate the AICc and RSSN with different model orders. The optimal model order is $N = 4$.

Then, the AICc criterion is applied to optimize the model order in this work (*Chang et al., 2018*). Equation (23) is the AICc criterion

$$AICc = T \ln RSSN + 2N + \frac{2N(N+1)}{T - N - 1} \qquad (23)$$

where $T$ is the number of samples, and $N$ the model order. $RSSN = \|Y - \Phi Z\|^2$ is the forecasting error. Here, when $T$ is small, the constraint ability of the AICc criterion on the number of parameters is strengthened, and so it is applicable to the case that the number of samples is small in this work.

Table 5 shows the performance indexes of the normal data and the data with the outlier with different model orders. Here, the number of groups is set to 4. It can be seen that, the performance indexes are smallest when the order $N = 4$, which indicates that the forecasting results are dependent on the model order. As shown in Table 6, the minimum

**Table 7 Comparison of performance indexes of different numbers of groups.**

| Group number | Performance indexes (no outlier) | | | Performance indexes (with outlier) | | |
|---|---|---|---|---|---|---|
| | RMSE (MW) | MAPE (%) | MAE (MW) | RMSE (MW) | MAPE (%) | MAE (MW) |
| 3 | 58.347 | 5.21 | 42.782 | 59.095 | 5.06 | 43.776 |
| 4 | 54.015 | 4.74 | 40.964 | 56.655 | 4.88 | 41.952 |
| 5 | 64.902 | 5.44 | 46.819 | 71.249 | 6.19 | 53.212 |
| 6 | 57.934 | 5.55 | 44.742 | 65.724 | 5.85 | 48.166 |
| 7 | 63.685 | 5.66 | 50.003 | 74.157 | 6.26 | 57.016 |

**Note:**
The data indicate the performance indexes of MCAR model with different groups. The forecasting error is minimum when the number of groups is 4.

value of the model order is 34.07 of AICc model. Meanwhile, the optimal model order is $N = 4$. This is consistent with the experimental result in Table 5.

Lastly, the number of groups in the MCAR model is also a variable parameter which can affect on the forecasting performance. Table 7 shows the performance indexes of MCAR model with different groups. Here, kernel width is optimized by the Silverman rule and the model order is fixed to 4. It can be seen that, whether or not the load data with the outlier, the forecasting error is minimum when the number of groups is 4 in this experiment. Here, when the number of groups is small, the model cannot find the similarity of load series. However, with the further increase of the number of groups, the similarity of the load series decreases, and the distribution difference of the time series becomes larger. It should be noted that, the optimal number of groups is different for different data sets. Thereby, in the actual modeling process, the number of groups needs to be selected according to the characteristics of the dataset.

## DISCUSSION

In this section, the performance of the proposed model is compared with some state of the art deep learning methods which have been applied for load forecasting, including adaptive recurrent neural networks (Adaptive RNN), long short term memory (LSTM) networks, gated recurrent units (GRU) and the combination model. The training samples in this section are taken from the power series from January to November, and the test samples from December. Settings of some parameters are shown in Table 8. To guarantee a fair comparison, the valuables of parameters of deep learning methods networks (Adaptive RNN, LSTM, and GRU) are set to same valuables, where these have been optimized. Here, the parameters of Adaptive RNN are set as recommended in *Fekri et al. (2021)*. The algorithm of LSTM is based on *Kong et al. (2019)*. The algorithm of GRU are based on *Li et al. (2020)*. Parameters of the combination model are set based on *Li & Chang (2018)*.

The performance indexes of the series with the outlier are tabulated in Table 9. The results show that, the proposed MCAR model displays promising results in terms of the average values of MAPE, MAE, and RMSE indexes for the series with the outlier, although LSTM produces less value for the normal data. As for the maximum values of MAPE, MAE, and RMSE indexes, the MCAR model also produces smaller values due to less sensitive to the outlier. Moreover, the MCAR model has less training times in comparison

**Table 8 Parameters setting of forecasting models.**

| Combinational model | Adaptive RNN, LSTM, GRU |
|---|---|
| Individual model: ARIMA, Elman, similarity model | Number of layers: 2 |
| Optimization algorithm: CPSO | Number of training samples: 8,760 |
| Knowledge capacity: 5 | Optimization algorithm: Adam |
| $\omega_{max}$, $\omega_{min}$: 2, 0 | Maximum number of iterations: 3,000 |
| $c_1$, $c_2$: 2.5, 0.5 | Number of neurons in the hidden layer: 50 |
| $m_C$, $m_I$: 2 | Learning rate: 0.01 |
| $n_C$, $n_I$: 5 | Batch size: 40 |
| Model order: 5 | Bidirectional: No |

**Table 9 Comparisons of performance indexes of studied forecasting models.**

| Performance indexes | | Forecasting models | | | | |
|---|---|---|---|---|---|---|
| | | Adaptive RNN | LSTM | GRU | Combination model | MCAR |
| RMSE (MW) | Averages | 79.562 | 77.683 | 79.006 | 81.329 | 56.655 |
| | Minimum | 1.587 | 1.492 | 1.556 | 1.920 | 1.620 |
| | Maximum | 150.041 | 149.478 | 153.275 | 154.451 | 107.358 |
| MAPE (%) | Averages | 6.625 | 6.442 | 6.840 | 7.424 | 4.880 |
| | Minimum | 0.123 | 0.097 | 0.088 | 0.111 | 0.079 |
| | Maximum | 9.072 | 8.743 | 9.518 | 11.389 | 7.961 |
| MAE (MW) | Averages | 58.669 | 56.504 | 59.774 | 61.502 | 41.952 |
| | Minimum | 1.590 | 1.511 | 1.685 | 2.227 | 1.365 |
| | Maximum | 94.074 | 93.258 | 93.662 | 118.468 | 79.347 |
| Training time (s) | – | | 746 | 767 | 253 | 96 | 26 |

Note:
The data show the performance indexes of the series with the outlier of some forecasting models. The proposed MCAR model displays promising results in terms of the average values of MAPE, MAE, and RMSE indexes.

with deep learning methods, since it only need several training samples and contrarily deep learning methods need a large number of training samples and accordingly more computing time.

Finally, the uniqueness and novelty of this work is introduced. The established MCAR model can automatically eliminate the influence of outliers without detecting outliers. The local similarity between the true value and the regression one is measured by the maximum correlation entropy, which reduces outlier in the optimization solution of regression model. The average value of MAPE can be reduced to 1.63% in comparison with some state of the art method (Adaptive RNN, LSTM, GRU, and combination model). However, the traditional methods need to use filtering methods, such as statistical learning methods and wavelet analysis, to detect outliers and set thresholds which are depended on the statistical characteristics of the series, and therefore have high time complexity.

## FINDINGS

The advantages and limitations of the proposed MCAR model are analyzed in this section. The model is a robust regression model that can automatically eliminate outlier interference. It does not need to detect outliers of time series and to set thresholds, which effectively improves the accuracy, efficiency and reliability. Therefore, it is suitable for online forecasting of time series disturbed by outliers and noises. Moreover, different from the deep learning model which requires a large number of training samples, the MCAR model only needs a small number of (number groups) data to train model, but it is unable to discover more information implied in the data set. Also, it is difficult to determine the optimal number of groups in the modeling process, and it can only be selected according to experience. How to optimize the number of groups is our further research content.

## CONCLUSIONS

This study has established a robust MCAR forecasting model for the time series with outlier. The local similarity between data is measured by the Gaussian kernel width of maximum correlation entropy. Therefore, the MCAR model reduces the sensitivity to the outlier and enhances the accuracy and robustness. The advantage is that there is no need to detect outlier and set thresholds. The average values of MAPE, MAE and RMSE indexes and the training time of MCAR model are decreased in comparison with deep learning methods. However, it should be pointed out that, external factors, such as weather and holidays, have not been considered in our model. In future, the forecasting model should incorporate the calendar, weather information, the temperature, workday and weekends to furthermore improve the forecasting performance. This is also our next research direction.

## ACKNOWLEDGEMENTS

The authors would like to thank Dr. Junli Liang, Professor of Northwestern Polytechnical University, for helpful advice.

### Funding

This work was supported by the National Natural Science Foundation of China (No. 61871318). The funders had no role in study design, data collection and analysis, decision to publish, or preparation of the manuscript.

### Grant Disclosures

The following grant information was disclosed by the authors:
National Natural Science Foundation of China: 61871318.

### Competing Interests

The authors declare that they have no competing interests.

## Author Contributions

- Jing Ren conceived and designed the experiments, performed the experiments, analyzed the data, performed the computation work, prepared figures and/or tables, authored or reviewed drafts of the article, and approved the final draft.
- Wei-Qin Li conceived and designed the experiments, analyzed the data, authored or reviewed drafts of the article, and approved the final draft.

## Data Availability

The data and the codes are available in the Supplemental Files.

## Supplemental Information

Supplemental information for this article can be found online at http://dx.doi.org/10.7717/peerj-cs.1251#supplemental-information.

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
