# Peer review of "A robust maximum correntropy forecasting model for time series with outliers"

_PeerJ Computer Science, doi:10.7717/peerj-cs.1251_

## Round 0.1 · original submission · Major Revisions

Based on the reviewers’ comments, you may resubmit the revised manuscript for further consideration. Please consider the reviewers’ comments carefully and submit a list of responses to the comments along with the revised manuscript.

Reviewer 1 ·

Basic reporting

1. Overall this is a well-written manuscript and I would agree that the professional English is used with professional figures and tables.
2. One comments is that more discussion about deep learning models can be added in the background discussion. One of their advantages is that weather and calendar information can be easily incorporated into the model input since the structure of neural networks is very flexible. More references can be found:
[1] Li Y, Zhang Y, Cai Y. A new hyper-parameter optimization method for power load forecast based on recurrent neural networks. Algorithms, 2021, 14(6): 163.
[2] Rajbhandari Y, Marahatta A, Ghimire B, et al. Impact study of temperature on the time series electricity demand of urban nepal for short-term load forecasting. Applied System Innovation, 2021, 4(3): 43.
[3] Jiang W. Deep learning based short‐term load forecasting incorporating calendar and weather information. Internet Technology Letters, 2022, 5(4): e383.

Experimental design

1. Overall the proposed method is well described and proven effective with comprehensive experiments. Two further questions are as follows to make the description more clear:
(1) How to judge whether outliers exist or not when applying the proposed method? Or the method can be proceeded without this concern?
(2) How to determine the model parameters for new problems? As in this manuscript, the optimal model parameters are chosen with trials and errors shown in the tables.

Validity of the findings

no comment.

Reviewer 2 ·

Basic reporting

1. The Abstract should be constructed in a concise manner that presents readers with an instructive map to the paper. Please compose your abstract in a logical and accurate reflection of the organizational structure of the paper. Your abstract reflects organizational structure of paper (i.e., presenting problem/focus of study, research questions, data sets or benchmark problems, methodology, findings, key points from discussion of findings, and implications/recommendations). The current one is long one, but is not well-arranged. Furthermore, the quantitative results should be clearly defined and reported in the abstract.
2. The technical contribution of this work is recommended that the author refines it by mentioning the arisen problem solution approach, containing the scope, the significance of the research, and the potential outcomes.
3. All variables should be written in italic. Some mathematical notations are not rigorous enough to correctly understand the contents of the paper. The authors are requested to recheck all the definition of variables and further clarify these equations.
4. Discussion section is comprehensive and well written; however, it would be better first that authors highlight their findings in the form of statements along with the conclusive data of statistical importance; mention how their findings are unique and novel; how these findings are in consensus with the existing values/ reports or how different are they from the already reported findings.
5. Clarifying the study’s limitations allows the readers to better understand under which conditions the results should be interpreted. A clear description of limitations of a study also shows that the researcher has a holistic understanding of his/her study. However, the authors fail to demonstrate this in their paper. The advantages and disadvantages of the proposed methodology should be discussed in the findings section. The benefits and drawbacks of the proposed method should be explained by the authors. What are the methodology(ies) and limitation(s) used in this work? Please describe the advantages in practice and the limitations of the research. The distinctions you made in your study on who, what, where, when, why, and how can be used to organize these limitations. There should be some explanation of the limitations of the proposed strategy in order to present an objective viewpoint and to have an unbiased view in the paper.
6. It is not clear if experimental results were obtained under the same experimental conditions. Are the simulations performed in the same situations? How do you guarantee a fair comparison? Add further details on how simulations were conducted.
7. The analysis and configurations of experiments should be presented in detail for reproducibility. It is convenient for other researchers to redo your experiments and this makes your work easy acceptance. A table with parameter setting for experimental results and analysis should be included in order to clearly describe them.
8. All of the values for the parameters of all algorithms selected for comparison are not given.
9. The conclusion section needs revisions. It should briefly describe the findings of the study and some more directions for further research. The authors should describe academic implications, major findings, shortcomings, and directions for future research in the conclusion section.

Experimental design

1. The technical contribution of this work is recommended that the author refines it by mentioning the arisen problem solution approach, containing the scope, the significance of the research, and the potential outcomes.
2. Discussion section is comprehensive and well written; however, it would be better first that authors highlight their findings in the form of statements along with the conclusive data of statistical importance; mention how their findings are unique and novel; how these findings are in consensus with the existing values/ reports or how different are they from the already reported findings.
3. Clarifying the study’s limitations allows the readers to better understand under which conditions the results should be interpreted. A clear description of limitations of a study also shows that the researcher has a holistic understanding of his/her study. However, the authors fail to demonstrate this in their paper. The advantages and disadvantages of the proposed methodology should be discussed in the findings section. The benefits and drawbacks of the proposed method should be explained by the authors. What are the methodology(ies) and limitation(s) used in this work? Please describe the advantages in practice and the limitations of the research. The distinctions you made in your study on who, what, where, when, why, and how can be used to organize these limitations. There should be some explanation of the limitations of the proposed strategy in order to present an objective viewpoint and to have an unbiased view in the paper.
4. It is not clear if experimental results were obtained under the same experimental conditions. Are the simulations performed in the same situations? How do you guarantee a fair comparison? Add further details on how simulations were conducted.
5. The analysis and configurations of experiments should be presented in detail for reproducibility. It is convenient for other researchers to redo your experiments and this makes your work easy acceptance. A table with parameter setting for experimental results and analysis should be included in order to clearly describe them.
6. All of the values for the parameters of all algorithms selected for comparison are not given.

Validity of the findings

1. The conclusion section needs revisions. It should briefly describe the findings of the study and some more directions for further research. The authors should describe academic implications, major findings, shortcomings, and directions for future research in the conclusion section.

---

## Round 0.2 · Minor Revisions

There are some minor revisions required. The paper can be re-submitted after these minor revisions.

Reviewer 1 ·

Basic reporting

no comment

Experimental design

no comment

Validity of the findings

no comment

Additional comments

Dear authors,
Thanks for revising and resubmitting the manuscript.

Reviewer 2 ·

Basic reporting

The current version of the paper presents an expressive improvement as compared to the previous one. The authors provided acceptable answers to all questions and no more issues were detected in the current manuscript. Therefore, this reviewer recommends the acceptance of the paper after minor revisions listed below:

1. Equations should be used with appropriate equation number. Do not use “…as follows…” “…is…”, “…expressed as…”, “…be written as…”, “…can be expressed as follows…”, etc.
2. 167th row on Page 11 should be corrected.
3. Organization of the paper presented as the last paragraph of the paper should be written with subsection numbers. “Lastly, a brief findings and conclusion is introduced.” should also be corrected.
4. Character size of some sentences in The Conclusion section should be adjusted.

Experimental design

The current version of the paper presents an expressive improvement as compared to the previous one. The authors provided acceptable answers to all questions and no more issues were detected in the current manuscript. Therefore, this reviewer recommends the acceptance of the paper.

Validity of the findings

The current version of the paper presents an expressive improvement as compared to the previous one. The authors provided acceptable answers to all questions and no more issues were detected in the current manuscript. Therefore, this reviewer recommends the acceptance of the paper.

Additional comments

The current version of the paper presents an expressive improvement as compared to the previous one. The authors provided acceptable answers to all questions and no more issues were detected in the current manuscript. Therefore, this reviewer recommends the acceptance of the paper after minor revisions listed below:

Minor edits:
1. Equations should be used with appropriate equation number. Do not use “…as follows…” “…is…”, “…expressed as…”, “…be written as…”, “…can be expressed as follows…”, etc.
2. 167th row on Page 11 should be corrected.
3. Organization of the paper presented as the last paragraph of the paper should be written with subsection numbers. “Lastly, a brief findings and conclusion is introduced.” should also be corrected.
4. Character size of some sentences in The Conclusion section should be adjusted.

---

## Round 0.3 · accepted · Accept

The revisions are satisfactory and the paper is recommended for publication.